# The Impact of Herbal Infusion Consumption on Oxidative Stress and Cancer: The Good, the Bad, the Misunderstood

**DOI:** 10.3390/molecules25184207

**Published:** 2020-09-14

**Authors:** Wamidh H. Talib, Israa A. AL-ataby, Asma Ismail Mahmod, Sajidah Jawarneh, Lina T. Al Kury, Intisar Hadi AL-Yasari

**Affiliations:** 1Department of Clinical Pharmacy and Therapeutic, Applied Science Private University, Amman 11931-166, Jordan; israa.adnan1985@yahoo.com (I.A.A.); asmamahmod1212@gmail.com (A.I.M.); sajidah.jawarneh@asu.edu.jo (S.J.); 2Department of Health Sciences, College of Natural and Health Sciences, Zayed University, Abu Dhabi 144534, UAE; Lina.AlKury@zu.ac.ae; 3Department of Genetic Engineering, College of Biotechnology, Al-Qasim Green University, Babylon 00964, Iraq; entesar@biotech.uoqasim.edu.iq

**Keywords:** antioxidant, natural products, anti-tumor, dietary agents

## Abstract

The release of reactive oxygen species (ROS) and oxidative stress is associated with the development of many ailments, including cardiovascular diseases, diabetes and cancer. The causal link between oxidative stress and cancer is well established and antioxidants are suggested as a protective mechanism against cancer development. Recently, an increase in the consumption of antioxidant supplements was observed globally. The main sources of these antioxidants include fruits, vegetables, and beverage. Herbal infusions are highly popular beverages consumed daily for different reasons. Studies showed the potent antioxidant effects of plants used in the preparation of some herbal infusions. Such herbal infusions represent an important source of antioxidants and can be used as a dietary protection against cancer. However, uncontrolled consumption of herbal infusions may cause toxicity and reduced antioxidant activity. In this review, eleven widely consumed herbal infusions were evaluated for their antioxidant capacities, anticancer potential and possible toxicity. These herbal infusions are highly popular and consumed as daily drinks in different countries. Studies discussed in this review will provide a solid ground for researchers to have better understanding of the use of herbal infusions to reduce oxidative stress and as protective supplements against cancer development.

## 1. Introduction

The intimate link between nutrition and health is well documented [1] and people in many countries have strong beliefs that foods provide more benefits than just being a source of energy [2]. Herbs and plant-derived natural products are considered the oldest medications in the world [3]. Plants were traditionally used to treat different ailments, including cancer, which is the second leading cause of death after cardiovascular diseases [4]. More than 30,000 plants were evaluated for their anticancer effects by the National Cancer Institute [5] and several studies were conducted to prove the anticancer potential of plants or their natural products [6,7,8,9]. However, the medical field shows very limited use of plants and dietary agents in cancer prevention and treatment.

The consumption of herbal infusions is very common in the Mediterranean region and globally. In a study conducted on 1260 cancer patients in Palestine, 60.9% were consuming herbs, mostly in the form of decoctions [10]. These drinks are mainly prepared from aromatic plants belonging to the following families: Lauracae, Umbelliferae, Lamiaceae, Myrtacae and Compositae [3]. The plants used in the preparation of herbal infusions were subjected to several studies and some of these plants exhibited potent antioxidant and anticancer properties. However, overconsumption of these herbal infusions may result in contradictory and side effects. This review summarizes the antioxidant capacities, anticancer potential, and possible toxicity of eleven widely-consumed herbal infusions.

### Oxidative Stress and Cancer

Oxidative stress is the unbalance between production and elimination of free radicals and reactive species, like reactive oxygen species (ROS) and reactive nitrogen species (RNS) [11,12]. Oxidative stress is responsible for causing damage in cells and vital biomolecules. It is associated with induction of chronic inflammation and subsequently the development of many diseases including cancer, diabetes, cardiovascular, neurological and pulmonary diseases [12]. Oxidative stress is produced by external sources, like UV radiation, toxic chemicals and drugs, by physiological changes such as aging and inflammation and by internal sources via enzymatic and non-enzymatic reactions [13,14]. Many enzymes play an essential role in oxidative stress formation. These enzymes include xanthine oxidase (XO), P450 complex, NADPH oxidase (NOX), uncoupled endothelial nitric oxide synthase (eNOS), arachidonic acid (AA), lipoxygenase, peroxisomes and cyclooxygenase (COX) [11,13]. On the other hand, superoxide radicals are non-enzymatically generated by mitochondrial respiration chain complex I (NADPH–ubiquinone oxidoreductase) and complex III (the ubiquinol–cytochrome c oxidoreductase) [11,13]. Complex III mediates ROS production which has significant effect in cancer development and progression [11]. Clearly, ROS interact and oxidize many cellular constituents involving proteins, lipids and nucleic acids followed reversibly or irreversibly with changes in the structure and the function of these molecules [15]. On the other side, inducible NOS produces significant amounts of RNS which play critical role in the induction of lipid peroxidation and consequently the production of other reactive species, like reactive aldehydes-malondialdehyde (MDA) and 4-hydroxynonenal (4-HNE) [12].

In our body, redox homeostasis balances the oxidative stress via enzymatic and non-enzymatic antioxidants. Several types of antioxidants have critical roles in ROS elimination, including dietary natural antioxidants, like tocopherol, selenium, β-carotene, ascorbic acid, polyphenol metabolites, and synthetic antioxidants (e.g., N-acetylcysteine). On the contrary, many endogenous antioxidant molecules contribute in this role, such as glutathione, α-lipoic acid, coenzyme Q, ferritin, uric acid, bilirubin, metallothionein, L-carnitine, and melatonin. Endogenous antioxidant enzymes are also involved in balancing the oxidative stress. These enzymes include superoxide dismutase (SOD), glutathione reductase (Gr), thioredoxin reductase (TRX), catalase (CAT), glutathione peroxidases (GPXs) and peroxiredoxins (PRXs) [13].

All cancer phases entailing initiation, promotion and progression are affected by oxidative stress. Oxidative stress has the ability to activate several transcription factors including nuclear factor (NF)-κB, hypoxia inducible factor (HIF)-1α, activator protein (AP)-1, p53, peroxisome proliferator-activated receptor (PPAR)-γ,β-catenin/Wnt, and nuclear factor erythroid 2-related factor 2 (Nrf2). These factors regulate the expression of diverse genes included in immune modulation, inflammatory response, carcinogenesis, metastasis, tissue remodeling and fibrosis [12,16]. Besides that, ROS activate signaling pathways associated with cell growth, e.g., p38MAPK, p70S6K, p90Rsk, JAK/STAT, JNK, ERK, RAS, AKT and phospholipase D [13]. Moreover, ROS can oxidize cysteine residues in tyrosine phosphatases, for example PTEN and PTP-1B, and decrease their activities. Such changes promote hyper-activation of the PI3K and AKT pathways [17]. Additionally, pro-angiogenic factors, like HIF-1, actuate the transcription of angiogenic factors, such as VEGF, leading to neovascularization [18]. NOX1-derived RO stimulates angiogenic switch in fibroblasts and matrix remodeling [19]. In addition, iron-induced oxidative stress by ferric nitrilotriacetate (Fe-NTA) assesses in p16/p15 tumor suppressor genes deletion which results in carcinogenesis [20].

Antioxidants in diet and supplements are widely used to protect cells from the damage induced by ROS. However, some clinical trials have shown conflicting results that do not support this concept. Based on recent studies, antioxidants may increase melanoma metastasis in mice [21,22] and accelerate tumor progression in later stages of lung cancer [23]. One explanation of the antioxidant activity in promoting tumor growth is the disruption of the ROS-p53 axis, which is related to the somatic mutation in p53 that occurred in the late stage of tumor progression [23]. In fact, mutant p53 isoforms cannot apply antioxidant activities, and rather induce intracellular ROS and promote a pro-tumorigenic survival [24]. Interestingly, the administration of mitochondria and non-mitochondria-targeted antioxidants resulted in two distinguished outcomes of liver cancer prevention by altering DNA repair [25]. Figure 1 describes the role of oxidative stress in cancer.

## 2. Herbal Infusions Antioxidant and Anticancer Capacities

Several plants with high antioxidant abilities and total phenolic contents have been screened out to be used as a rich source of natural antioxidants. These plants could be developed into herbal infusions, functional food or pharmaceuticals for the inhibition and treatment of diseases caused by oxidative stress [26]. Medicinal plants with potent anticancer activities might be potential sources of vigorous natural antioxidants and beneficial chemopreventive agents [27]. In this section, eleven popular herbal infusions are evaluated for their antioxidant and anticancer beneficial effects.

### 2.1. Lemon and Ginger Combination

Lemon (*Citrus limonum*) belongs to the Rutaceae family. GC-MS/MS analysis showed the presence of high concentrations of limonene with percentages in the lemon water extract of 23.271% [6]. Limonene protected cells against the oxidative stress induced by the exogenous addition of H_2_O_2_. Limonene also defended normal lymphocytes from diseases related to oxidative stress, including cancer [28]. Moreover, (+)-limonene epoxide enhanced the activity of antioxidant enzymes like catalase and superoxide dismutase in mice [29]. Lemon peel essential oil exhibited 55.09% inhibition of 2,2-diphenyl-1picrylhydrazy l (DPPH), while ascorbic acid (positive control) showed a 5.18% activity, demonstrating its potent antioxidant effects [30]. It was found that peels of citrus fruits are a significant source of various antioxidants, and such by-products of the juice extraction industry could be utilized as natural antioxidants. Using the whole extract instead of individual antioxidants allows taking advantage of additive and synergistic impacts of diverse phenolics, flavonoids, ascorbic acid, carotenoids, and reducing sugars present in the samples [31].

Many indolofuroquinoxaline derivatives in lemon citrus have displayed promising growth prevention effect against the K562, MDA-MB 231, and MCF7 cell lines, though, no significant effects have been seen on the HEK293 cell line (normal cells), suggesting a selectivity of these derivatives towards cancer cells [32]. Studies have shown that lemon and grapefruit peel essential oils showed moderated to weak cytotoxicity against the human prostate (PC-3), lung (A549), and breast (MCF-7) cancer cell lines [33]. In lemon juice, the presence of limonene together with other components like alkaloids, phenols, flavonoids, and terpenoids was shown to be responsible for inducing apoptosis and inhibiting angiogenesis in cancer cells [6].

Ginger (*Zingiber officinale* Roscoe) belongs to the Zingiberaceae family. Herbal teas prepared from ginger are used as a folk remedy to treat coughs, colds, and flu. It is also applied as a paste for external applications to treat headaches [34]. Gingerols and shogaols are considered significant ingredients in ginger, as both of these ingredients exhibit biological activities, including anticancer, antioxidant, antimicrobial, anti-inflammatory, and anti-allergic properties [35]. Ginger has antioxidant properties and there is a positive relationship between antioxidant activities and total phenolic contents in ginger [36]. The antioxidant capacity of ginger infusion was measured by using spectroscopic methods and the result were 16.0 μmol gallic acid equivalent per gram of ginger extract [37]. Ginger can improve hepatic changes after an administration of a high dose of acetaminophen in vivo. This hepatic protection is caused by reduction of oxidative stress and increase in antioxidant capacity [38].

Oleoresin, extracted from the ginger’s rhizomes contains [6]-gingerol which suppresses cell adhesion, invasion, motility, and activities of MMP-2 and MMP-9 in the MDA-MB-231 human breast cancer cell line [39]. The cancer preventive properties of ginger have also been linked to the presence of flavonoids and polyphenolic components, particularly quercetin [40]. The fresh, dried, and steamed ginger has an antiproliferative effect against human Hela cancer cells. Interesitingly, the antiproliferative effect of steamed ginger at 120 °C for 4 h was found to be approximately 1.5- and 2-fold higher than that of dried and fresh ginger [41].

### 2.2. Wild Thyme (Thymus Serpyllum)

*Thymus serpyllum* belongs to the Lamiaceae family. It is a perennial shrub that has a woody base [42]. Rosmarinic acid is the principal ingredient identified in aqueous tea infusion (93.13 mg/g) of wild thyme. Rosmarinic acid possesses a variety of biological features, including anti-oxidant, anti-inflammatory, anti-viral, and anti-bacterial effects [43].The strong protective impact of wild thyme infusions is proposed to be the consequence of large amounts of rosmarinic acid and flavonoids (quercetin, eriocitrin, luteolin-7-*O*-glucoside, apigenin-7-*O*-glucoside, luteolin, apigenin) [44]. Wild thyme is a good source of compounds essential to prevent oxidation of low-density lipoproteins in vivo [43]. By using the pFRAP method, wild thyme infusion extracted for 30 min showed significantly the highest average antioxidant activity (268.01 mg GAE/100 g), while extraction for 60 min showed lowest antioxidant activity (111.56 mg GAE/100 g). It is concluded that the antioxidant activity of the tea of wild thyme depends on extraction time [45]. Another study conducted by Zhang et al. showed that rosmarinic acid in thyme increases the activity of superoxide dismutase, catalase, and glutathione peroxidase with a reduction in malondialdehyde [46]. Moreover, wild thyme essential oils demonstrated better overall antioxidant activity compared to other thymus species, due to the presence of thymol in its essential oil [47].

In osteosarcoma cells, rosmarinic acid showed anticancer effects by suppressing DJ-1 via regulation of the PTEN-PI3K-Akt signaling pathway [48]. Previous study confirmed that rosmarinic acid reverses non-small cell lung cancer cisplatin resistance through the activation of the MAPK signaling pathway [49]. Moreover, rosmarinic acid inhibited lung metastasis of murine colon carcinoma cells by activating AMP-activated protein kinase [50]. The hexane extract of Thymus serpyllum was cytotoxic to six different cancer cell lines.The highest anticancer activity was found in HepG2 (Liver Carcinoma Cell Line), followed by HCT 116 (a colon cancer cell line), MCF7 (breast cancer cell line), MDA-MB-231 (breast cancer cell line), PC3 (prostate cancer cell line), and A549 (lung carcinoma cell line) [51].

### 2.3. Marjoram (Organum Majorana)

*Organum majorana* belongs to the Lamiaceae family. Marjoram is a shrub and a perennial plant native in Asia and the Mediterranean area. It has been used traditionally in the folk medicine as an antifungal, antiviral, and antiparasitic remedy [52]. Using the active oxygen method [53] and ferric reducing antooxidant properties [54], methanolic extracts of marjoram have shown potent antioxidant activity that was attributed to the presence of polyphenolic compounds in the plant. Up to date, 31 polyphenols were identified in marjoram [55]. Recently, LC–ESI-MS/MS analysis detected rosmarinic acid as the most potent antioxidant polyphenol in marjoram’s methanolic extract [56], while gas chromatography-mass spectroscopy analysis of essential oil of both the stem and aerial parts revealed the presence of linalool and estragole as main components [57]. The marjoram water extract plays an important role in the initiation of apoptosis by inducing DNA damage in human colon cancer HT-29 cells and down-regulation of survivin (inhibitor of apoptosis) and the activation of caspases, in human breast cancer MDA-MB-231 cells [58]. Supporting this finding, the essential oil of marjoram showed cytotoxic anti-cancer effect against the HT29 and Caco-2 colon cancer cell lines, partially through the down-regulation of survivin [59]. The pure essential oil of marjoram was also shown to cause a concentration- and time-dependent reduction in the proliferation of the lung cancer cells (A549 and LNM35) and the growth of their relevant colonies in vitro. Likewise, treatment with marjoram significantly reduced the growth of LNM35 and A549 xenografts in the chick embryo and in nude mice models in vivo without notable side effects [60]. Moreover, the highest phenolic contents and antioxidant actions of the marjoram water extract lead to the upregulation of cyclin-dependent kinase inhibitor 1 (p21), leading to apoptosis and suppression of the cell cycle in the breast cancer MCF-7 cell line [61].

### 2.4. Palestinian Herbal Mix

The Palestinian herbal infusion contains green tea, lemon verbena, sage, and citrus lemon. Green tea (*Camellia Sinensis*) belongs to the Theaceae family and it is widely consumed in Asian countries [62]. Fresh tea leaves contain caffeine, theobromine, theophylline, and other methylxanthines, lignin, organic acids, chloro-phylland, theanine, and free amino acids [63]. Moreover, other components exist, including, flavones, phenolic acids, and depsides, carbohydrates, alkaloids, minerals, vitamins, and enzymes [64]. Tea polyphenols, essentially flavonoids, are well-known for their antioxidant capacities. Various studies have confirmed that polyphenols and tea catechins are exceptional electron donors and efficient scavengers of physiologically-relevant ROS in vitro, including superoxide anions [65]. The most critical bioactive agent in green tea is epigallocatechin-3-gallate [66], which is listed as an antioxidant. Epigallocatechin 3-gallate exerts its beneficial biological actions directly by interacting with proteins and phospholipids in the plasma membrane and regulating signal transduction pathways, transcription factors, DNA methylation, mitochondrial function, and autophagy [67]. Green tea, which is rich in polyphenols, has been found to increase the inhibitory effect of tamoxifen on the proliferation of the ER (estrogen receptor)-positive MCF-7, ZR75, and T47D human breast cancer cells in vitro [68]. The dietary green tea polyphenol has a potentiating impact on cisplatin anti-tumor activity and a protective influence against cisplatin-induced renal dysfunction. It is suggested that green tea polyphenol may be used with cisplatin as a modulator in anticancer treatment [69]. Interestingly, the grape extracts work synergistically with decaffeinated green tea extracts in the prevention of the activity of tumor-associated NADH oxidase (tNOX) and the prevention of cancer cell growth. Intra-tumoral injections of 25:1 mixture of the green tea extract and ground freeze-dried pomace were effective in repressing the growth of 4T1 mammary tumors in mice [70]. Combining (-)-epigallocatechin gallate and quercetin synergistically prevented stem cell characteristics of human prostate cancer cells [71].

Lemon verbena (*Lippia citriodora*) belongs to the Verbenaceae family. It is a shrub with scented leaves that grows in both tropical and subtropical regions [72]. Chemical composition of lemon verbena showed that it is composed of large amounts of polyphenolic compounds including verbascoside (400 mg/L) and luteolin 7-diglucuronide (100 mg/L). It also contains 42 mg/L of essential oil with much more citral (77% of the essential oil) [73]. Infusion of *Lippia citriodora* protected against lipid peroxidation and protein carbonylation. Also, the decoction showed higher antioxidant capacity compared to the infusion [74]. Leaf infusion of lemon verbena worked as a free radical scavenger and exhibited an antigenotoxic activity by increasing the antioxidant status [75]. In a clinical trial including 43 healthy subjects, lemon verbena leaves induced oxidant/antioxidant balance by causing a reduction in the lipid peroxidation and an increase in the total antioxidant ability [76]. Lemon verbena essential oil showed high antiproliferative activity against a panel of human cancer cell lines (A375 > Caco2 > HepG2 > MCF-7 > THP-1). Citral or geranial is the main component in the essential oil of lemon verbena, and while it has strong anticancer and antimicrobial properties, it has weak direct antioxidant activities [77]. Lemon verbena is also rich with luteolin, which is a flavone bioflavonoid and has anticancer properties. It induces apoptotic cell death, inhibits the proliferation of cancer cells, and inhibits tumor angiogenesis [78].

Sage (*Salvia officinalis*) belong to the Lamiaceae family. Sage is a perennial shrub, an aromatic and remedial plant endemic in the Mediterranean region [79]. According to Gas Chromatography-Mass Spectrometry (GC-MS) analysis, the main detected compounds were oxygenated monoterpenes followed monohydrocarbone, squiterpenes, and others. The main essential oil constituents were α-terpineol (33.07%), camphor (11.57%), α-pinene (8.96%) camphene (5.09%) β-cymene (5.40%) caryphyllene (3.76%) β-myrcene (3.65%) β-menthen-1-ol (3.45%) and bomeol (3.38%) [80]. Several in vivo and in vitro studies have investigated the activity of polyphenols as sage tea active ingredients that may prevent lipid peroxidation and augment antioxidant defense mechanisms [81]. Daily drinking of sage in both mice and rat causes significant increase in the liver antioxidant enzyme glutathione-S-trans- ferase (GST) activity [82]. Similar effect was observed when rat hepatocytes (isolated from the livers of sage drinking rats) showed an increase in glutathione (GSH) level and GST activity. Also, the treatment of rats with water extracts of sage for five weeks protected rat hepatocytes against azathioprine toxicity [83]. It is important to note that the method of extraction affects the antioxidant activity. The highest antioxidant activities of sage were discovered in the methanolic extract, followed by water infusion and decoction [84]. The hydroalcoholic extract of sage exhibited high antioxidant activity [85]. For example, earlier study has shown that the hydroalcoholic extract of sage has hepatoprotective action against isoniazid-induced hepatic damage in rats. This activity may be attributed to free radical-scavenging, and antioxidant activities of flavonoids in the extract [86]. Furthermore, many diterpenes, isolated from plants of several species of the genus Salvia, have been demonstrated to have interesting antitumor activity [87]. S. officinalis essential oil inhibited human HNSCC cell by activating different anticancer mechanisms [88].

### 2.5. Lebanese Herbal Mix

This infusion contains green tea, lemon verbena, cinnamon, damask rose, chamomile flowers, primrose, and ginger. Cinnamon (*Cinnamomum zeylanicum*) belongs to the Lauraceaeis family. It is a small evergreen tree with a height of 5–7 m. The plant is charecterized by the aromatic odor and pleasant smell. The biological activities of cinnamon are due to the presence of tannins [89]. The bark of C. zeylanicum essential oil was analyzed by GC–MS which revealed the presence of (*E*)-cinnamaldehyde (68.95%), as the major component, in addition to benzaldehyde (9.94%), and (*E*)-cinnamyl acetate (7.44%) [90]. Saponins, tannins, phenols, terpenoids, and phytosterols were observed in the cinnamon plant whether dried or fresh [91]. Cinnamon’s essential oil prevents the hepatic 3-hydroxy-3-methylglutaryl CoA reductase activity in rats which leads to lower hepatic cholesterol content and decrease the lipid peroxidation via enhancement of the hepatic antioxidant enzyme activities [92]. Differences in antioxidant activities of cinnamon maybe related to different parts of the plant. For example, cinnamon leaf oils have high antioxidant activities, whereas cinnamon bark oils have low antioxidant activities [93]. A cinnamon water extract contained the highest amount of phenolics and had the highest antioxidative activity [94]. Earlier study has investigated the hepatoprotective activity of both aqueous and ethanolic extracts of cinnamon against carbon tetrachloride (CCl_4_) that induced lipid peroxidation and hepatic injury in rats. The raised serum AST and ALT enzymatic activities induced by CCl_4_ were significantly decreased by oral administration of 200 mg/kg of each extract once daily for seven days, as compared to the untreated rats [95].

A review of the literature showed that cinnamon has various cytotoxic activities against different cancer cell lines, namely basal cell carcinoma, human epithelioid cervix carcinoma (HeLa), human cancer promyelocytic leukemia (HL-60), human colorectal carcinoma (HCT 116, HT 29, and SW 480), epidermoid carcinoma (A431), and human cervical carcinoma (SiHa) [96]. The alcoholic and aqueous extracts of the stem bark of *Cinnamomum malabatrum* possess protective effects against Dalton’s Ascitic lymphoma-induced cancer in mice. Such activity is due to the presence of flavonoids, essential oils, amino acids, tannins, and phytosterols [97]. At a concentration of 1.28 mg/mL (including 10.24 µM cinnamaldehyde), aqueous cinnamon extract treatment resulted in 35–85% growth prevention of the majority of the cancerous cells. Similarly, a concentration of 10 µM cinnamaldehyde treatment resulted in a 30% growth prevention of only SK-N-MC cells with no effect on other cancer cell lines. These results suggest that aqueous cinnamon extract had a significant inhibitory effect on the majority of cancer cells [98].

Damask rose (*Rosa damascena* Mil.) belongs to the Rosacea family. It is a deciduous shrub and is the most significant aromatic medicinal plant. The damask rose is mainly used in the perfume industry and as a flavoring agent in food products [99]. In *R. damascena* essential oil, a total of 22 compounds were detected by GC-MS analysis. Both citronellol (23.43%) and geraniol (34.91%) were the main scent compounds of the fresh rose flowers [100]. The entire flavonoid content of aqueous and ethanolic extracts of *R. damascena* flower petals were found to be 12.73% and 32%, respectively. These flavonoids work as potent antioxidant agents [101]. Moreover, the main volatile component of rose water is geraniol (3.3–6.6%); meanwhile, in the essential oil, the geraniol component represents 8.3–30.2% [102]. The rose oil showed a remarkable inhibition against acetylcholinesterase (60.86 ± 1.99%) and butyrylcholinesterase (51.08 ± 1.70%) at 1000 μg/mL and moderate activity in DPPH radical scavenging and ferric reducing antioxidant power tests [103]. Leaf methanolic extracts (hot and cold) of *R. damascena* displayed anti-free radical activity at a concentration of 50 μg/mL. The leaf cold extraction had the most potent antioxidant activity measured with the FRAP assay at the concentration of 100 μg/mL, compared with the hot methanolic extraction [104]. Fresh and spent *R. damascena* flower extracts showed 74.51% and 75.94% antiradical activities, while the antioxidant activity of fresh flower extract (372.26 mg/g) was higher than that of spent flower extract (351.36 mg/g) [105]. The damask rose essential oil was found to have significant cytotoxic impacts against the cancer cell line (A549) in comparison with the normal cell line (NIH3T3) [106].

Chamomile (*Matricaria chamomilla* L.) belongs to the Asteraceae family. It is a slow growing aromatic annual plant with branched stems, double feathery shared leaves, and tiny, soft, hollow, lettuce head flowers [107]. The foremost components of chamomile include the terpenoids α-bisabolol and its oxides and azulenes, including chamazulene [108]. GC-MS and GC-FID analysis discovered the qualitative and quantitative composition of the chamomile flowers essential oil. The achieved results revealed the presence of 52 components, and the essential contents were β-farnesene (29.8%), α-farnesene (9.3%), α-bisabolol and its oxide (15.7%) and chamazulene (6.4%).

The antioxidant activity of *M. chamomilla* was investigated using DPPH assay [107]. The antioxidant effect of water and alcohol extracts of chamomile flowers on long-term storage of anhydrous butter fat was measured by peroxide value and free fatty acids. The results of this study revealed a moderate effect in controlling hydrolytic rancidity. However, the antioxidant effect of the water extract was described to be significantly higher than the alcohol extract [109]. The antioxidant activity and stability were investigated with three methods, DPPH free radical scavenging system, determination of the peroxide, in addition to thiobarbituric acid numbers. The antioxidant activity of the essential oil was evaluated in 0.2, 0.4, 0.6, 0.8, and 1 mg/mL concentrations by measuring peroxide and thiobarbituric acid numbers in crude sunflower oil as a greasy food. The antioxidant activities of the extracts were increased by increasing the extract concentrations [110]. The highest total phenolic content and maximum antioxidant capacity of aqueous extract of chamomile tested at temperatures 25, 80, and 100 °C were achieved at 80 °C. For the aqueous herbal extract, total phenolic content was significantly correlated with antioxidant activity [111]. In a single-blind randomized controlled clinical trial conducted on 64 subjects (males and females; age between 30 to 60 years), the antioxidant capacity, superoxide dismutase, glutathione peroxidase, and catalase activities were significantly (*p* < 0.05) increased by 6.81%, 26.16%, 36.71%, and 45.06%, respectively in chamomile group compared with patients in the control group [112].

Primrose (*Primula vulgaris*) belongs to the Primulaceae family. The primrose flower is funnel-shaped, with orange spots at the base of its lobes [113]. *p*-coumaric acid and rutin have been recognized as the main phenolics in primrose water extract [114]. Primrose reduced H_2_O_2_-induced DNA damage in a concentration-dependent manner in fibroblast cells compared to the positive controls (only 20 μM H_2_O_2_ treatment) [114]. Using MTT assay, dimethyl sulfoxide extract of primrose flowers showed a cytotoxic effect on lung (A549), liver (HepG2), breast (MCF-7), and prostate (PC-3) cancer cells [115]. Further, the extract exhibited selective cytotoxic impacts against human cervical cancer cells (HeLa cells) by arresting their cell cycle at the S phase [116].

A recent study on five herbal infusions including lemon and ginger combination, wild thyme, marjoram, Palestinian and Lebanese herbal mix was conducted to evaluate their antcancer activities. The water extract of lemon and ginger combination was found to be the most potent against MDA-MB231, MCF-7, and A549 cell lines. Both lemon and ginger combination and wild thyme separately, showed the highest apoptosis induction and angiogenesis suppression abilities on the MDA-MB231 cell line at concentrations of 3.5 and 4.4 mg/mL, respectively. Furthermore, lemon and ginger combination, wild thyme, marjoram, and the Lebanese herbal drink were the most active extracts in stimulating pinocytosis, respectively while Palestinian herbal drink had a moderate effect [117].

### 2.6. Roselle (Hibiscus sabdariffa L.f)

Roselle (*Hibiscus sabdariffa* L.) is a well-known species that belongs to the Malvaceae family. The plant is an annual or perennial herb found in tropical and subtropical regions of the world. It is used to produce phloem fibers and as an infusion (herbal tea) [118,119]. A preliminary phytochemical analysis has shown the presence of alkaloids, tannins, saponins, glycosides, phenols, and flavonoids in different solvent extracts of *H. sabdariffa* [120]. Studies have also shown that calyces of *H. sabdariffa* contained alkaloids, flavonoids, saponins, tannins, and a high anthocyanin content with the lowest flavonoids and phenolic acid [121]. Gas chromatography/mass spectrometry (GC-MS) analysis has identified 18 volatile components in the calyx of *H. sabdariffa*, most of which were fatty acids and ester compounds [122]. Ethanimidic acid and its ethyl ester (31%) were the major phytocompounds detected in the methanol extract of hibiscus flowers, which are reported to possess antioxidant and cancer-preventive properties [123]. A previous study has reported the presence of anthocyanins in *H. sabdariffa* calyces in two major forms; delphinidine-3-sambubioside and cyanidine-3-sambubioside [124]. The relation between total anthocyanin content (TAC) and antioxidant activity of hibiscus infusion has been investigated and revealed that under optimum conditions (10 g/mL, 88.7 °C and 15.5 min), TAC was 132.7 ± 7.8 mg and antioxidant activity was high according to DPPH and ABTS assays [125]. The antioxidant effect of anthocyanins is due to their phenolic structure that has the ability to scavenge ROS [126]. Extraction conditions, like temperature, extraction time, and solid to solvent ratio, affect the antioxidant activities of *H. sabdariffa*, which tends to be high when the extraction temperature is in the range of 70–80 °C and the extraction time is from 120 to 150 min. At the same time, it decreases as the solvent to solid ratio is increased [127]. Furthermore, bound soluble phenolic compounds in *H. sabdariffa* extracts and their impact on antioxidant activity have been investigated and have shown that the highest total phenolic content (TPC) resulted in high antioxidant scavenging capacity [128]. In a later study, *H. sabdariffa* aqueous extract exhibited a higher ability to scavenge peroxyl radicals in the water environment than in the lipophilic system and exhibited a more potent metal-reducing activity than olive leaf extract [129].

Previous studies have shown that *H. sadbariffa* is a promising anticancer plant against different cancer types. The aqueous extract of *H. sadbariffa* was able to reduce cell viability and induced apoptosis of human adenocarcinoma cell line (MCF-7) [130]. In this regard, a study has indicated that anthocyanin-rich extract from *H. sadbariffa* calyx was able to inhibit tumor growth, lung metastasis, and tumor angiogenesis. All these effects could be mediated via inhibition of tumor Ras, NF-κB, CD31, and VEGF/VEGF-R-induced angiogenesis [131]. In a recent study, cytotoxic and antitumor activities of polyphenolic leaf extract of H. sabdariffa were investigated. It has shown the ability of the extract to reduce the growth of breast cancer cell lines (MDA-MB-231) and estrogen receptor-expressing breast cancer cell lines (MCF-7 and T-47-D) [132]. H. sabdariffa aqueous flower extract was able to induce apoptosis in a human gastric carcinoma cell line (AGS) via the JNK/p38 signaling cascade [133].

### 2.7. Pomegranate (Punica granatum)

Pomegranate (*Punica granatum* L.) is a member of the Punicaceae family and is considered to be of Middle East origin. The plant has also been used in the traditional medicine for ages [134]. *P. granatum* is a good source of phenolic compounds and anthocyanins, as demonstrated in a previous study [135]. A preliminary phytochemical screening has revealed the presence of different phytochemicals from ethanolic, aqueous, and chloroform extracts of pomegranate peel, whole fruit, and seeds. The extracts of the entire fruit contained the majority of the phytocompounds represented by triterpenoids, steroids, glycosides, saponins, alkaloids, flavonoids, tannins, carbohydrates, and vitamin C. [136]. *P. granatum* peels ethanolic extract has been analyzed using GC-MS chromatography and showed various constituents with antioxidant activity such as decahydro-1-pentadactyl- naphthalene, 5-hydroxymethyl furfural and 1, 3-cyclohexadiene [137]. According to a recent study, two extracts of *P. granatum*—hydroalcoholic and infusion—exhibited high efficiency in inhibiting DPPH radical and had significant reducing power of the Fe^3+^/ferricyanide complex. The antioxidant activity of fruit peel extracts has been justified by the presence of a high level of phenolic compounds like ellagic acid and its derivatives, as identified by UPLC-PDA-MS analysis [138]. Moreover, a study carried on arils juice and peel decoction of fifteen varieties of *P. granatum* and has shown that the TPC and total flavonoids content (TFC) of arils juices were about 20- and 300-fold inferior to decoctions. Regardless of variety, each decoction revealed better antioxidant and chelating activity compared to the juices [139]. The antioxidant activity of *P. granatum* peel was assessed using three aqueous extraction techniques; continuous shaking extraction, maceration, and hot water infusion. It was found that infusion method resulted in significant (*p* < 0.05) level of antioxidant activity compared to other extraction techniques [140]. Since *P. granatum* has an impact on ROS, an in vivo study was carried out to test the effect of methanolic extracts of pomegranate seeds and peel on oxidative stress induced by methotrexate. The results have demonstrated a significant reduction in GPX and SOD, and an improvement in MDA values after methotrexate treatment [141].

The use of *P. granatum* preparations has a long ethnomedical history and preclinical studies have described different pharmacological abilities, including chemopreventive, chemosensitization, and chemotherapeutic activities [142]. It was reported that *P. granatum* has the ability to down-regulate various signaling pathways like NF-ᴋB, P13K/AKT/mTOR and Wnt, as well as reduece the expression of genes that are associated with cancer development, such as anti-apoptotic genes, MMPs, VEGF, c-met, pro-inflammatory cytokines, cyclines, and Cdks [143]. In this regard, the antitumor effect of punicalagin, a pomegranate polyphenol, has been investigated in a human prostate cancer cell line (PC-3) and LNCaP cells. It was found that punicalagin inhibited cell viability of both cell lines in a dose-dependent manner and induced the expression of caspase-3 and -8 in PC-3 [144]. Moreover, an in vivo study has shown that aqueous extract of pomegranate fruit has anticancer activity against Ehrlich-ascites-carcinoma (EAC)-bearing Swiss albino mice. After intraperitoneal injection of mice with pomegranate aqueous extract, a significant reduction has occurred in tumor volume and weight, as well as, decreased viable cell count and improvement in the life span of EAC bearing mice [145]. In another study, pomegranate fruit juice and two of its components (ellagic acid and luteolin) have been shown to reduce cell viability of ovarian cancer cell line (A2780), inhibit metastasis via down-regulation of matrix metalloproteinases MMP2 and MMP9. Also, all three treatments inhibited tumor growth of ovarian cancer cell line (ES-2) in nude mice experiments [146].

### 2.8. Anise Seeds (Pimpinella anisum L.)

*Pimpinella anisum* L. is an annual herbaceous plant that belongs to the family Apiaceae [147]. It is widely used as a flavoring agent and as a primary ingredient in herbal infusions [148]. A recent study was conducted to evaluate the antioxidant potentials of Portuguese *P. anisum* seeds infusion. The results of this study have detected high content of flavonoids, phenols, and anthocyanins. These bioactive components reflected the high antioxidant activity of *P. anisum* against free radicals [149]. Use of a combination of anion-exchange, gel filtration, and hydrophobic interaction column chromatographies facilitated the isolation of three lignin-carbohydrate protein complexes from a hot water extract of the seeds of *P. anisum* [150]. The results of GS-MS analysis of *P. anisum* infusion have shown the presence of fatty acids (linoleic, oleic, and palmitic acids), triterpenoids (lupeol, β-amyrin and betulinic acids), and sterols (β-sitosterol and stigmasterol) [149]. Anethole is the major constituent of anise seed oil [151].

A previous study has demonstrated the correlation between total phenolic content and antioxidant activity of both anise and cumin infusions (ground form). It was found that phenolic compounds in these infusions are the main contributors to their free radical-scavenging activity and oxidant reduction potency [152]. Another study has investigated two extracts—water and alcohol—of chamomile flowers, anise seeds and dill seeds. The extracts exhibited significant antioxidant activity in both linoleic acid and liposome model systems, however the water extract showed higher activity comparing to alcohol extracts [148]. Moreover, anise aqueous extract was assessed in streptozotocin-induced diabetic rat model and showed pancreatic damage reduction via modulation of insulin secretion, oxidative stress, autophagy and down-regulation of caspase 3 [153].

Consumption of anise seed has many medicinal benefits, such as anticancer, hepato-protective and antioxidant abilities. In this regard, a comparative study with cisplatin has conducted to evaluate the cytotoxic effect of (PA) aqueous extract on oral squamous cell carcinoma (KB cell line). It was reported that anise seed extract exhibited anticancer activity by being able to reduce cell viability in dose dependent manner [154]. Moreover, *P. anisum* extracts and essential oil have shown antiproliferative effect on gastric cancer cells (AGS), and anti-angiogenesis activity in HUVEC cells [155]. Another study has investigated the cytotoxic effect of anise seed ethanolic extract on human prostate cancer cell line (PC-3). The treatment with *P. anisum* extract showed significant anticancer activity comparing to the normal cell line [156].

### 2.9. Cumin (Cuminum cyminum)

Cumin (*Cuminum cyminum* L.) is a small annual herbaceous plant that belongs to the Apiaceae family [157]. It is originated from Egypt, Turkistan, Iran, and Eastern Mediterranean [158]. The bioactive phytochemicals found in cumin seeds are associated with their various industrial applications that range from food to pharmaceutical products [157]. Phytochemical analysis of *C. cyminum* has revealed the presence of alkaloids, anthraquinones, coumarins, flavonoids, glycosides, resins, saponins, tannins and steroids [159]. Cumin seed essential oil has been analyzed using GC-MS analysis and the presence of 18 compounds demonstrated, with 3-caren-10-al and cuminal being the main constituents [160]. A previous study has investigated the antioxidant activity of alcoholic and aqueous extracts of cumin seeds. Results have shown that the alcoholic extract had higher activity comparing to the aqueous extract [161]. It was found that alcoholic extracts of *C. cyminum* have a higher phenolic content and antioxidant capacity comparing to coriander extracts [162]. Cumin seed aqueous extract was able to protect WRL-68 cells from hexavalent chromium-induced oxidative injury via reducing ROS in dose-dependent manner. The antioxidant potential of the cumin seed extract is positively correlated with the high content of phenolic acid [163].

The antioxidant activity of cumin seeds plays a role in its cytotoxic ability against the human cervical carcinoma HeLa cell line. It was reported that at concentration of 0.1 μl/mL, essential oil of cumin reduced HeLa cells by 79% [164]. Another study investigated the activity of ethanolic extract of *C. cyminum* L. against seven human cancer cell lines and showed 61% maximum cytotoxic activity present in Colon 502713 cell line [165]. Various cumin seed extracts exhibited anticancer and neuro-protective effects against IMR32 human neuroblastoma cell lines [166]. Different flavonoids have been purified and identified from *C. cyminum* and showed anticancer potency against breast cancer MCF-7 cell line [167].

### 2.10. Lemon Balm (Mellissa officinalis L.)

*Mellissa officinalis* L. is popularly known as lemon balm and belongs to Lamiaceae family. It is an edible perennial herb that has been used for ages in the form of decoctions, infusions or a natural flavoring in food [168,169]. Although distributed worldwide, lemon balm is originated from Asia and Europe [170]. The chemical composition of *M. officinalis* oil has been analyzed using the GC-FID technique. The major components were geranial (34%) and neral (26%) [171,172,173]. Several studies have demonstrated a high content of phenolic compounds in lemon balm aqueous extract [172,174,175]. The most abundant phenolic compound found in lemon balm was rosmarinic acid (derived from caffeic acid), as well as some flavonoids such as luteolin-7-*O*-glucoside [172,176]. *M. officinalis* exhibited strong antioxidant activity, which were 10 times stronger than the antioxidant effects of vitamin C and vitamin B [177]. An in vivo study has shown the efficacy of *M. officinalis* aqueous extract in reducing Mn-induced brain oxidative stress in mice [178]. Furthermore, antioxidant properties of four Lamiaceae species have shown that *M. officinalis* has the highest total phenolic content and antioxidant activity compared to the other species [173]. In this regard, lemon balm extract and its major constituent, rosmarinic acid, effectively attenuated the oxidative stress by inducing antioxidant enzymes and alleviating liver damage in an animal model of nonalcoholic steatohepatitis [179]. Another study has also reported that *M. officinalis* extracts have strong antioxidant capacity and DPPH radical scavenging activities compared to butylated hydroxytoluene (BHT) [180].

The anticancer potency of lemon balm has been previously studied using different types of extracts on various tumor cell lines [181,182]. Aqueous extract of *M. officinalis* has shown chemo-preventive effect against hepatocellular carcinoma (HCC) in rats and exhibited antioxidant activity via increasing GSH concentration and inhibiting lipid peroxidation in the liver tissues of HCC rats [183]. Lemon balm extracts inhibited cancer progression and angiogenesis in the ovo CAM model with high cell inhibitory against MCF-7 breast cancer cell line [184]. Moreover, five different extracts of *M. officinalis* have shown cytotoxic activity against three human tumor cell lines: NCI-H460 (non-small cell lung cancer), MCF-7 (breast adenocarcinoma), and AGS (gastric adenocarcinoma) [174].

### 2.11. Rosemary (Rosmarinus officinalis L.)

Rosemary (*Rosmarinus officinalis* L.) belongs to the mint family Lamiaceae which is widely distributed in the Mediterranean region [185,186]. This plant has phenolic diterpenes and triterpenes as the main active constituents, namely caffeic acid, rosmarinic acid (RA), ursolic acid, carnosic acid, and carnosol, all of which are reported for their antioxidant activities [187,188]. Essential oil of rosemary contains α-pinene (45.7%), camphene (18.3%), eucalyptol (16.9%) and p-cymene (6.4%), berbonone, in addition to camphor bornyl acetate [187]. Steam distillated rosemary oils contain mainly 1,8-cineole (46.4%), camphor (11.4%) and α-pinene (11.0%), camphor (37.6%), 1,8-cineole (10.0%), p-cymene-7-ol (7.8%) and borneol (5.4%). Moreover, R. officinalis L. (RO) is rich in flavonoids that have antioxidant activities such as 30-*O*-β-d-glucuronide, 7-*O*-glucoside, hispidulin, diosmin, genkwanin, hesperidin and isoscutellarein 7-*O*-glucoside which are found in flowers, leaves, roots and stems of *R. officinalis* [188]. Along with the antioxidant activities, *R. officinalis* has anti-inflammatory, hepatoprotective, antidiabetic and antimicrobial activity which depend on the content of the phenolic compounds mainly rosmarinic acid, caffeic acid and carnosic acid [189,190,191,192].

In view of the antioxidant and antiproliferative activities of R. officinalis, many in vitro studies confirmed these effects depending on the polyphenolic content. For example, Ðilas et al. investigated the effect of a number of oil-soluble rosemary extracts with varying content of carnosic acid, carnosol and methylcarnosol. Results of this study revealed that the extract with the highest content of carnosic acid had the most powerful scavenging ability to hydroxyl, superoxide and 2,2-diphenyl-1-picrylhydrazyl (DPPH)-free radicals. Tested rosemary extracts also exhibited significant antiproliferative effect indifferent cell lines. In both breast adenocarcinoma (MCF7) and cervix epitheloid carcinoma (HeLa) celll lines, the extracts yielded low IC_50_ (9–10 μgmL^−1^) [193]. Supporting this finding, rosemary extract inhibited MDA-MB-231 breast cancer cell proliferation, prevented the phosphorylation/activation of Akt and mTOR and enhanced the cleavage of PARP, a marker of apoptosis, indicating that rosemary extract modulates key signaling molecules involved in cell proliferation and survival [194]. Pro-apoptotic effect of rosemary crude extract (ursolic acid, carnosol and carnosic acid) was also indicated in 184-B5/HER cells via increased sub-G_0_ cell population along with suppression of G_1_-S phase transition and reduction of cyclin D1 expression. As a result, the colonies of 184-B5/HER cells were reduced significantly [195].

The antiproliferative effect of rosemary extract and its active ingradients carnosol, carnosic acid and rosmarinic acid on human ovarian cancer cells was investigated by Tai et al. Rosmary extract was shown to have significant antiproliferative effect in human ovarian cancer cells (A2780) and cisplatin resistance daughter cell line A2780CP70 with effective IC_50_ at 1/1000 and 1/400 dilutions, respectively. Moreover, the extract and its active ingradients enhanced the antiproliferative effect (synergistic effect) of cisplatin in both A2780 and A2780CP70 cells. The effect of rosemary extract was due to the suppression of the expression of Bcl-2, Bcl-x, cIAP-1, HIF-, and HO-1 (anti-apoptosis proteins) and Bax, Fas and FADD (pro-apoptosis molecules). Furthermore, A2780 cells were more sensitive to carnosol, carnosic acid and rosmarinic acid than A2780CP70 cells. Interestingly, rosmary extract upregulated a heme-containing protein, cytochrome C, that was emitted from the mitochondria in response to pro-apoptotic stimuli [196]. Alternatively, rosemary extract inhibited cell proliferation and enhanced apoptosis of human NSCLC adenocarcinoma A549 cells in a dose-dependent manner. This effect was associated with the reduction in phosphorylated/activated Akt, mTOR and p70S6K levels [197]. Additionally, aqueous extract of the fruits of rosemary showed anti-cancer and cytotoxic effect against gastrointestinal cell lines, KYSE30 (human esophageal squamous cell carcinoma) and AGS (human gastric carcinoma). Using MTT assay, IC_50_ value was 150 mg/mL after 72 h of exposure in KYSE30 cell lines, however, in AGS cell lines, IC_50_ value was 1.3 mg/mL after 72 h of exposure [198]. In prostate cancer, rosmarinic acid minimized cell proliferation through the downregulation of p53 expression [199].

In vivo, rosemary polyphenols have antiproliferative effects against colon cancer cells in animal models. For example, when male nude mice were grafted with human colon cancer cells (HT-29) treated with fluid extract of rosemary, the extract exhibited a clear reduction in the tumor size. This outcome was strongly correlated with the role of rosemary extract in sharp increase of intracellular ROS that stimulated necrotic cell death. In addition, Nrf2 gene silencing increased rosemary cytotoxic effects [200]. Another study revealed that administration of 200 mg/kg of rosemary extract per day in mice leads to the reduction in the progression of colorectal cancer HT-29 cells. This is attributed to the alteration in RNA post-transcriptional modification, protein synthesis and the amino acid metabolism which are responsible for tumor reduction. For example, rosemary extract altered nucleic acid binding capacity, followed by enzyme modulator proteins, hydrolases, cytoskeletal proteins, oxidoreductases and transferases [201]. In another nude mouse model grafted with HCT116 cells and treated with rosemary extract and carnosic acid, the extract was also able to inhibit the HCT116 xenograft tumor initiation [202]. Table 1, summarizes the all discussed herbs and their active ingredients, mechanism of action to reduce oxidative stress and the types of cancers that are tested.

## 3. Herbal Infusions in Human Clinical Trials

A short-term intake of chamomile tea (3 g/150 mL hot water, three times daily) has beneficial effects on glycemic control and antioxidant status in patients with type 2 diabetes mellitus [112]. Consumption of green tea extract for six weeks combined with CrossFit training resulted in a significant increase in the blood antioxidant capacity and a marginal effect on aerobic capacity and serum brain-derived neurotrophic factor in trained men [231]. Moreover, a randomized double-blinded, placebo-controlled phase II clinical trial was conducted to evaluate the effect of green tea extract (GTE) on mammographic density (MD). Results showed a reduction in percent MD in younger women (50–55 years), but with no significant effect on MD measures in all women [232].

A randomized controlled trial suggested that aromatherapy (1 mL of lemon and 0.5 mL of ginger essential oil) has efficacy in preventing treatment-related salivary gland disorder [233]. Interestingly, a pilot study’s outcomes have shown that administering 2 g of ginger daily in patients with high risk for colorectal cancer may improve cell-cycle biomarkers in the normal-appearing colonic mucosa [234].

Cumin essential oil supplementation enhanced some antioxidative indices, as superoxide dismutase and total antioxidant capacity in patients with metabolic syndrome [235]. Based on a randomized crossover trial, pomegranate juice consumption for eight weeks in hemodialysis patients resulted in beneficial effects on blood pressure, oxidative stress, inflammation, and serum lipoprotein cholesterol [236]. Moreover, pomegranate extract (900 mg daily) intake by colorectal cancer patients has moderate modulation of specific tissue microRNAs (colorectal cancer biomarkers) [237]. Studies have shown that supplementation with pomegranate juice significantly affects oxidative stress by improving antioxidant response [238,239]. The oral administration of *Melissa officinalis* infusion markedly improved oxidative stress conditions and DNA damage that arises from radiation exposure among radiology staff [240].

## 4. Herbal Infusion Contradictory Effects

Altough herbal infusions are widely consumed, some studies have reported a lack of effect or even contraductory responses. For example, one of the common uses of lemon juice was the treatment of high blood pressure. Quercetin, a flavonoid present in citrus lemon, was found to lower lower blood pressure in several rat models of hypertension [241]. Interestingly, in 2012, a study showed that lemon juice has no beneficial effect on raised blood pressure, despite the common usage of lemon juice by hypertensive patients. In addition, lemon juice has potential risks for hypertensive patients like drug interaction and noncompliance with prescribed treatments. For example, it has been demonstrated that lime juice inhibits activity of cytochrome P450 3A4 [242].

In contrast to a previous study which showed that ginger interferes with iron absorption [243], a recent study concluded that ginger improves iron absorption, and therefore, it is beneficial as a supplement in the therapy of anemia [244].

*H. sabdariffa* has been used in the traditional medicine as an anti-hyperlipidemic agent [245]. Randomized conrolleed trials have demonstrated that *H. sabdariffa* has no significant role to lowering serum lipids when compared with placebo, black tea, or diet [246]. Alternatively, recent study of randomized clinical trials assessed the effect of *H. sabdariffa* on fasting plasma glucose (FPG), total cholesterol (TC), high-density lipoprotein (HDL), low-density lipoproteins (LDL), and triglyceride (TG). It was shown that *H. sabdariffa* has no significant effect on blood TC, HDL, and TG, however, *H. sabdariffa* was able to lower both FPG and LDL [247].

Despite the potential activity of phytochemicals as therapeutics for chemoprevention, they have certain limitations that are related to the complex mixture of metabolites, biphasic effects (hormesis), and their bioavailability [248]. Hormesis is known as a biphasic dose-response phenomenon such that a chemical has a stimulatory effect at low doses but is toxic at high doses. It is charecterized by either a U-shaped or an inverted U-shaped dose-response curve, based on the end-point measured [249]. Many phytochemicals are described as hormetins due to their hermetic response [250]. For example, *P. granatum* seed oil exhibited a hormetic effect when applied to mouse mammary organ culture. This was evident by the high inhibitory effect on on tumor at low tested concentration [251]. Moreover, the inactivity of green tea polyphenols against superoxide free radicals may be due to high doses used, which resulted in a pro-oxidant toxic effect [252]. The main constituent of green tea, (−)-epigallocatechin-3-gallate, revealed biphasic dose-response on a broad range of cell types (non-tumor and tumor cell lines) [253].

In addition to the abovementioned effects, thymol exhibited a hormetic effect in terms of antioxidant and pro-antioxidant activity, cell viability and DNA genotoxicity [254]. Thyme essential oil improved the metabolic activity of MCF-7 cancer cell line only at low concentrations [255]. In MCF-7 and HeLa cell lines, both lemon balm Kombucha and tea exhibited biphasic response at different concentrations [256]. Figure 2 shows the role of each herb in fighting cancer through oxidative stress pathways.

## 5. Conclusions

Herbal infusions are important sources of antioxidant agents and can be used to reduce oxidative stress and protect against cancer. Diverse phytochemicals are present in these infusions with the ability to activate different mechanisms involved in lowering oxidative stress and enhancing anticancer effects. Extraction temperature and time are important variables for some herbal infusions in order to obtain the highest antioxidant effect. However, uncontrolled consumption of herbal infusions may cause toxicity and reduced antioxidant activity.

## Figures and Tables

**Figure 1 molecules-25-04207-f001:**
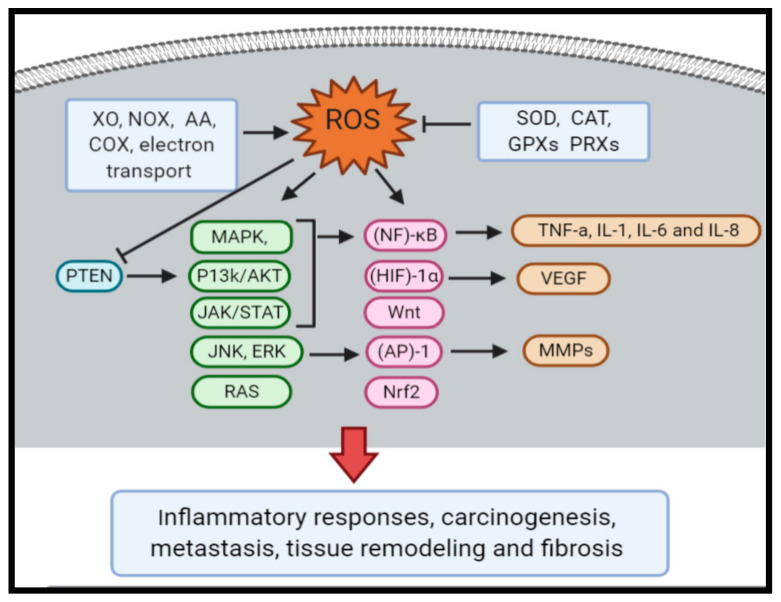
The role of the oxidative stress in cancer. ROS are generated by enzymes such as xanthine oxidase (XO), NADPH oxidase (NOX), nitric oxide synthases (NOS), arachidonic acid (AA) and cyclooxygenase (COX) and by mitochondrial respiration chain, this production is countered by endogenous antioxidant enzymes (e.g., superoxide dismutase (SOD), catalase (CAT), glutathione peroxidases (GPXs) and peroxiredoxins (PRXs). However, over production of ROS activate several transcription factors including nuclear factor (NF)-κB, hypoxia inducible factor (HIF)-1α, activator protein (AP)-1, p53, Wnt, and Nuclear factor erythroid 2-related factor 2 (Nrf2), which regulate the expression of genes included in inflammatory responses, carcinogenesis and metastasis, tissue remodeling and fibrosis. Furthermore, ROS activate signaling pathways associated with cell growth, e.g., JAK/STAT, JNK, ERK, RAS and AKT. Moreover, ROS oxidize cysteine residues in phosphatase and tensin homolog (PTEN) and decrease their activities, hence, these changes reveal activation of the PI3K/AKT pathways. Also, HIF-1 actuate transcription of angiogenic factors, such as Vascular endothelial growth factor (VEGF), leading to neovascularization. In addition, NOX1-derived ROS upregulate VEGF, VEGF receptors and matrix metalloproteinases (MMPs).

**Figure 2 molecules-25-04207-f002:**
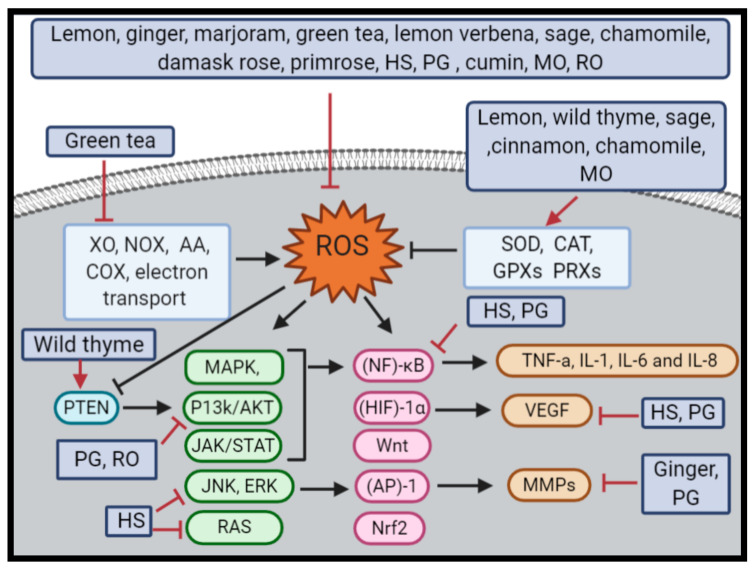
The role of each herb in fighting cancer through oxidative stress pathways. ROS and free radicals are reduced by lemon, ginger, marjoram, green tea, lemon verbena, sage, chamomile, damask rose, primrose, *Hibiscus sabdariffa* L. (HS), *Punica granatum* (PG), cumin, *Mellissa officinalis* L. (MO) and rosemary (RO). Moreover, antioxidant enzymes are increased via lemon, wild thyme, sage, cinnamon, chamomile and MO, on the other hand, green tea reduce ROS-inducing enzymes. Clearly, HS and PG reduce the activity of (NF)-κB, which induce the proinflammatory cytokines (e.g., TNF-a, IL-1, IL-6 and IL-8), and they repress VEGF, besides PG inhibition activity on P13k/AKT pathway and HS inhibition activity on JNK and RAS pathways. As observed, ginger and PG paly important role on MMPs inhibition. Wild thyme increases the activity of PTEN, which consequently reduce P13k/AKT pathway. Finally, RO have the ability to scavenge free radicals and have a role in prevention of Akt.

**Table 1 molecules-25-04207-t001:** Antioxidant and anticancer activities of the phytochemical components in the eleven herbal infusions.

Name of the Herbal Infusion/Ref.	Extracts/Oils	Active Ingredients	Antioxidant and Anti-Tumor Mechanisms	Type of Cancer Treated	Cell Lines Used (In Vitro)
Lemon [203,204]	water extract, volatile oils, lemon juice	limonene, ascorbic acid, phenolics, flavonoids, carotenoids, reducing sugars, indolofuroquinoxaline, alkaloids, terpenoids, geranial, neral	reduced exogenous H_2_O_2_ effect, enhance the activity of catalase and SOD, inhibit DPPH, decrease the expression of BcL-2 and the proliferative marker Ki-67, downregulate of caspase 3	myeloid leukemia, prostate, lung and breast, gastric cancer	K562 MDA-MB 231 MCF7 PC-3 A549 AGS BGC-823 SGC-7901
Ginger [205,206]	aqueous extract, oil/water soluble extract	gingerols and shogaols, gallic acid, quercetin	reduce oxidative stress and raise total antioxidant capacity, represse activities of MMP-2 and MMP-9, increase p53, CASP2 and DEDD, high expression levels of ABCA2 or ABCA3 transporter genes	breast and cervical cancer, ovarian, leukemia	Hela MDA-MB-231 SKOV-3 CCRF-CEM Nalm-6
Wild thyme (*Thymus serpyllum*) [207,208]	aqueous extract, essential oils, hexane extract	rosmarinic acid, eriocitrin, luteolin, apigenin, quercetin, luteolin7-*O*-glucoside, apigenin-7-*O*-glucoside, luteolin, apigenin, thymol, p-cymene, caryophyllene camphene eucalyptol and β-pinene	prevent oxidation of low-density lipoproteins, increases the activity of SOD, catalase, and GPXs, reduce malondialdehyde, reduce DJ-1 via regulation of the PTEN-PI3K-Akt signaling pathway, activate MAPK signaling pathway and AMP-activated protein kinase, decrease of cells in the S phase	liver carcinoma, colon, breast, prostate and lung, pancreatic cancer, osteosarcoma, melanoma	MDA-MB-231 MCF-7 HepG2 HCT-116 PC3 A549 PANC-1 U2OS A375 B164A5
Marjoram (*Organum Majorana*) [209,210]	methanolic extracts, water extract, essential oil, ethanolic extract aqueous extract	rosmarinic, linalool, estragole	reduce ferric reducing ability, down-regulation of survivin, upregulation of cyclin-dependent kinase inhibitor 1 (p21), activate caspase-dependent extrinsic apoptotic pathway and TNFα pathway, suppress NF-kB	breast and lung cancer, colon, liver cancer	Caco-LNM35 A549 MDA-MB231 MCF-7 HT29 HepG2
Green tea [211]		caffeine, theobromine, theophylline, lignin, organic acids, chloro-phylland, theanine, free amino acids, depsides, carbohydrates, alkaloids, minerals, vitamins, enzymes, polyphenols, tea catechins, epigallocatechin-3gallate, polyphenols, quercetin, epigallocatechin gallate	electron donors and efficient scavengers, interact with proteins and phospholipids in the plasma membrane and regulates signal transduction pathways, transcription factors, DNA methylation, mitochondrial function, and autophagy, prevents of tNOX activity, modulate Bax/blc-2 ratio and trigger G2/M cell cycle arrest	breast cancer, non-small lung cancer	MCF-7 ZR75 T47D A-549
Lemon verbena [212,213]	crude extract	verbascoside, luteolin 7-diglucuronide, citral or geranial, luteolin, verbascoside, gardoside	protected against lipid peroxidation and protein carbonylation, free radical scavenger, increase in the total antioxidant ability, modulate AMPK activity, decrease NF-κB, increase GST and GPx	human melanoma, human leukemia, colon, liver, brest cancer	A375 Caco2 HepG2 MCF-7 THP-1
Sage [214,215]	water extracts, essential oil, methanolic extract, hydroalcoholic extract, n-hexane soluble extract	were α-terpineol, camphor, α-pinene, camphene, β-cymen, caryphyllene, β-myrcene, β-menth1-en-b-ol, bomeol, flavonoids, diterpenes, manool	prevent lipid peroxidation, increase in the liver antioxidant enzyme GST activity, increase in glutathione (GSH) level and free radical-scavenging	head and neck squamous cell carcinoma, Hodgkin lymphoma, melanoma, human breast cervical, human hepatocellular carcinoma, MO59J, U343 and human glioblastoma, lung.	HNSCC L-540 HD-MyZ HepG2 MO59J U343 U251 NCI-H187
Cinnamon [216,217]	essential oil water extract aqueous and ethanolic extracts, distillate oil	(E)-cinnamaldehyde, benzaldehyde, (E)-cinnamyl acetate, saponins, tannins, phenols, terpenoids, and phytosterols, flavonoids and amino acids, coumarin, melatonin	decrease the lipid peroxidation via enhancement of the hepatic antioxidant enzyme activities	basal cell carcinoma, cervix carcinomacancer, leukemia, colorectal carcinoma, epidermoid carcinoma, brain cancer, breast cancer	HeLa HL-60 HCT-116 HT-29 SW-480 A431 SiHa SK-N-MC MCF-7, MDA-MB-231, BT-549
Damask rose [218]	essential oil, aqueous and ethanolic extracts, methanolic extracts	flavonoid, citronellol, n-nonadecane, n-heneicosane, 1-nonadecene, geraniol	inhibits acetylcholinesterase and butyrylcholinesterase, radical scavenging and ferric reducing antioxidant	lung	A549
Chamomile [219,220]	water and alcohol extracts, methanol extract, hydroalcoholic Extract	terpenoids α-bisabolol and its oxides and azulenes, including chamazulene, β-farnesene, α-farnesene, α-bisabolol, and its oxide and chamazulene, bisabololoxide A	free radical scavenging, increase SOD, GPXs, and catalase activities, reduce lipid peroxidation	Leukemia, colon	K562 HT29
Primrose [221,222]	water extract, dimethyl sulfoxide Extract, oil, crude aqueous ethanolic extract	ρ-coumaric acid and rutin, decane, campesterol, caryophyllene, sitosterol, flavanol (proanthocyanidins)	reduces H_2_O_2_-induced DNA damage, increases malondialdehyde, and TNF-α, decrease NF-kB, cyclooxygenase-2, and MMP -9	lung, liver, breast, and prostate and cervix cancer cells, colon cancer	A549 HepG2 MCF-7 PC-3 HeLa SW-480
*Hibiscus sabdariffa* L. [223,224]	methanol extract, aqueous extract, ethanolic extract, n-hexane extract, ethyl acetate extract	alkaloids, tannins, saponins, glycosides, flavonoids (anthocyanin), alkaloids, phenolic acid, ethanimidic acid and ethyl ester	scavenge ROS and free radicals, potent metal-reducing activity, inhibits tumor Ras, NF-κB, CD31, and VEGF/VEGF-R-induced angiogenesis, JNK/p38 signaling cascade -induced apoptosis, increase activation of p21, p53, and caspase-3	adenocarcinoma, breast cancer, estrogen receptor-expressing breast cancer, human gastric carcinoma, lung cancer	MCF-7 MDA-MB-231 T-47-D A549
Pomegranate [225,226]	alcoholic, aqueous, chloroform extracts, juice	anthocyanins, triterpenoids, steroids, glycosides, saponins, alkaloids, flavonoids, tannins, carbohydrates, and vitamin C, naphthalene, decahydro-1-pentadactyl, 5 hydroxymethyl furfurals, and 1, 3-cyclohexadiene, ellagic acid and luteolin, polyphenols	scavenger for free radical and significant reducing power of the Fe^3+^/ferricyanide complex, down-regulate various signaling pathways like NF-κB, P13K/AKT/mTOR, and Wnt, reduces MMPs, VEGF, c-met, pro-inflammatory cytokines, cyclines, and Cdks, induces the expression of caspase-3 and -8, reduce phosphorylation levels of Akt, S6K1, inhibit IGF-I/Akt/mTOR pahway	prostate cancer, Ehrlich-ascites-carcinoma and ovarian cancer, thyroid cancer	PC-3 LNCaP A2780 ES-2 DU145 BCPAP
Anise seeds (*Pimpinella anisum* L.) [227,228]	water extract, alcohol extract, ethanolic extract, aqueous-n-butanolic extract, essential oils	flavonoids, phenols, and anthocyanins, lignin-carbohydrate protein, fatty acids (linoleic, oleic, and palmitic acids), triterpenoids (lupeol, β-amyrin and betulinic acids), and sterols (β-sitosterol and stigmasterol), anethole, gallic Acid, catechins, estragole, naringin, chloroginic acid, rosmarinic acid	scavenge DPPH free radicals, reduce oxidant potency, down-regulate of caspase 3	oral squamous cell carcinoma, gastric cancer, human prostate cancer, breast cancer	AG HUVEC PC-3 MCF-7
Cumin (*Cuminum cyminum* L.) [163,229]	essential oil, alcoholic and aqueous extracts	alkaloids, anthraquinones, coumarins, flavonoids, glycosides, resins, saponins, tannins, steroids, 3-caren-10-al and cuminal	reducing ROS, diminish the expressions of mTOR and survivin and elevate BECN1 expression	cervical, colon cancer, neuroblastoma, breast cancer	Hela 502713 IMR32 MCF-7 AU565
*Mellissa officinalis* L. [182,230]	essential oil, aqueous extract, infusion extracts, hydromethanolic, hydroethanolic, methanolic and alcoholic extracts, dichloromethane extract	geranial and neral, luteolin-7-*O*-glucoside, caffeic acid, 3,4-dihydroxyphenyl lactic acid, 3,4-dihydroxybenzoic acid, lithospermic acid, luteolin-7-*O*-glucoside, methyl caffeate and rosmarinic acid.	inducing antioxidant enzymes and alleviated liver damage, free radical scavenging activities, increasing GSH concentration and inhibiting lipid peroxidation in the liver tissues, reduce pro-caspase 3 levels	liver, non-small cell lung cancer, breast and gastric cancer	HCC MDA-MB-231 MCF-7 NCI-H460 AGS LNCAP, PC3 MDA-MB-468
Rosemary [191,202]	oil-soluble extracts, aqueous extract, essintial oils, crude extract	caffeic acid, rosmarinic acid, ursolic acid, carnosic acid, and carnosol, α-pinene, camphene, eucalyptol, p-cymene, camphor bornyl acetate, berbonone, 1,8-cineole	scavenge DPPH, prevent Akt and mTOR, reduce cyclin D1, suppress expression of Bcl-2, Bcl-x, cIAP-1, HIF-, and HO-1, Bax, Fas and FADD	breast, cervical, colon, ovarian, lung, esophageal and prostate cancer	MCF7 HeLa A2780 A2780CP7 A549 MDA-MB-231 KYSE30 184-B5/HER HT-29 HCT116 HDAC2

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
