# Peer review of "The Impact of Herbal Infusion Consumption on Oxidative Stress and Cancer: The Good, the Bad, the Misunderstood"

_molecules, 2020, doi:10.3390/molecules25184207_

Round 1

Reviewer 1 Report

In this manuscript, Talib et al., reviewed the Impact of Herbal Infusions consumption on Oxidative Stress and Cancer. The work is interesting. However, there are some concerns from this reviewer.

Most of the cited work were from in vitro cell culture not from in vivo models. You need to check if any human clinical trial data are available to support your conclusions.

You may focus on ~5 popular infusions and discuss more on mechanisms of action.

The whole review base on a principle “antioxidants are good”, but remember there are many publications found antioxidants promote tumor growth (after tumor is formed).

Moreover, phenolic compounds possess hometic effects on carcinogenesis, there are U-shaped effects (see numerous reviews from Calabrese et al.,)

Check your headings after 2.4 Palestinian herbal mix, you start with “3.4. Lebanese herbal mix”.

Section 3. Herbal infusion toxicity: Do you really need this section? It is OK for a Food Toxicology Journal, but not for Food Chemistry. It is rare to have toxic effect from drinking herbal tea. It is also not common to see someone eat ginger at equivalence to 66.67 g/kg/day. The dose makes the poison!

Author Response

Thank you for your comments

Attached are a detailed response for your comments.

Regards

Reviewer 2 Report

[Molecules] Manuscript ID: molecules-914705 - Review

Title: The Impact of Herbal Infusions consumption on Oxidative Stress and Cancer: The Good, the Bad, the Misunderstood

Journal: Molecules

In my opinion the review article is not suitable for publication in the current version, as it must be better organized and better written.

The review article entitled “The Impact of Herbal Infusions consumption on Oxidative Stress and Cancer: The Good, the Bad, the Misunderstood” is a good attempt to summarize the impact of herbal infusions on oxidative stress and cancer. The authors of the manuscript have described all the aspects of 10 herbs very comprehensively which is very good to read and gives a very broad uses and applications of this wonderful herb/aromatic plant. Though the manuscript is comprehensive and many new aspects were discussed severe requirement for English language improvement. Although this article is good but needs major revision throughout the manuscript.

General:

Please Add a list of abbreviations.

The Key words are repeated into the title: herbal infusions, natural products, oxidative stress, cancer, dietary antioxidant. Please change some of the keywords to be different the title

Insert line numbers into the text in order to do the review.. it’s very difficult to do a good review with this format

Please insert information about Rosemary. It is a very important herb with effects on oxidative stress and cancer.

Table 1 shows a summary for each herb and its extracts, active ingredients, mechanism of action to fight oxidative stress and the types of cancers that are tested by this herb. Please add more information if they are studies in vivo, in vitro.. more details about the study.. etc. I would recommend a column on it and also to show how these compound in herb are useful for oxidative stress and cancers and reference them with researchers who have first worked on it. Also the references are quite old, please introduce the latest references about this field.

Finally, I suggest a higher attention into writing scientific names, because often they are not reported properly (please add the genus with capital letter and both genus and species in italic)

The format is not Molecules format. Please change format according the journal. See author instruction.

Concluding, my suggestion is to re-organize the whole manuscript, maybe focusing deeply on the aspect you would like to emphasize, avoiding general statements if not strictly necessary, and then re-submit to the journal.

Author Response

Thank you for your comments

Attached is a detailed response to your comments.

Regards

Round 2

Reviewer 1 Report

I can recommend to accept this revised version for publication.

Reviewer 2 Report

Thank you for the modifications. The manuscript has improved after the review. In my opinion, the manuscript could be accepted in present form.